# Testing the Use of Standardized Laboratory Tests to Infer Hg Bioaccumulation in Indigenous Benthic Organisms of Lake Maggiore (NW Italy)

**Davide A.L. Vignati** [1,2,*] **, Roberta Bettinetti** [3] **, Angela Boggero** [4] **and Sara Valsecchi** [2]

1    Université de Lorraine, CNRS, LIEC, F-57000 Metz, France
2    National Research Council-Water Research Institute (CNR-IRSA), Via Mulino 19, 20861 Brugherio, MB, Italy; valsecchi@irsa.cnr.it
3    University of Insubria, DiSUIT, Via Valleggio 11, 22100 Como, Italy; roberta.bettinetti@uninsubria.it
4    National Research Council-Water Research Institute (CNR-IRSA), Corso Tonolli 50, 28922 Verbania, Italy; angela.boggero@irsa.cnr.it
*    Correspondence: david-anselmo.vignati@univ-lorraine.fr

**Abstract:** The chronic toxicity of mercury essentially derives from its strong tendency to biomagnify along food webs. For this reason, the European regulatory framework establishes an environmental quality standard for Hg based on the total Hg concentration in prey fish to protect top predators. A considerable part of the Hg burden of prey fish can come from the ingestion of benthic organisms that, in the presence of contaminated sediments, may remobilize substantial amounts of Hg towards the pelagic food webs. The present study evaluated whether Hg accumulation in assemblages of indigenous chironomids and oligochaetes could be predicted using standardized laboratory bioaccumulation tests with *Chironomus riparius* and *Lumbriculus variegatus*. Indigenous chironomids and oligochaetes were recovered at different sites in a lake suffering from legacy Hg pollution and analyzed for total Hg content. Sediment aliquots from the same sites were used to assess Hg bioaccumulation using laboratory-reared *C. riparius* and *L. variegatus*. Mercury concentrations in indigenous versus laboratory organisms showed a good correlation ($p < 0.05$; Spearman correlation test) only in the case of *C. riparius* versus indigenous chironomids, suggesting the possibility of using linear regressions to predict Hg accumulation by these benthic invertebrates. Further research needs and caveats as to the applicability of the present results to other aquatic systems are identified and discussed.

**Keywords:** mercury; sediments; bioaccumulation; chironomids; oligochaetes

---

## 1. Introduction

Mercury is a persistent environmental contaminant that can severely threaten human and environmental health. Because of its peculiar environmental chemistry, characterized by the presence of a gaseous species (Hg(0)) and of organic complexes (mainly monomethylmercury—MMHg), mercury is ubiquitous and tends to bioaccumulate and biomagnify strongly in food webs [1]. Although Hg is a neurotoxin and one of the most (eco)toxic trace elements, its chronic toxicity is linked essentially to dietary uptake [2]. Protection of top predators from secondary poisoning via ingestion of contaminated preys must, therefore, be the primary focus of Hg risk management. These specific management needs are reflected, *inter alia*, in the current European regulation that establishes an Environmental Quality Standard for Hg based on concentration in prey fish [3]. However, as much as 65% of the diet of non-piscivorous fish can come, directly or indirectly, from benthic sources with sediment-dwelling organisms representing a crucial link between primary producers, detrital deposits

and top predators [4,5]. Indeed, mercury concentrations at the base of the food web, rather than variation in biomagnification rates, seem to be the main factor controlling Hg transfer in food webs [6,7].

Considering that sediments represent a major repository for many persistent chemicals introduced into aquatic systems, it is important to understand to what extent sediment-bound Hg is potentially available for remobilization via benthic organisms [8]. Indeed, from a risk-assessment point of view, mercury requires an approach based on establishing its bioaccumulative potential in resident biota rather than the typical toxicity-based approach commonly used for other trace elements [9,10]. Data from low trophic levels (i.e., plankton and benthic invertebrates) are necessary to estimate the trophic magnification factors (TMF) for a given food web to fulfil e.g., European regulatory requirements for bioaccumulative chemicals [11]. In the case of small organisms such as freshwater benthic invertebrates, collecting an adequate amount of biomass of resident biota for measuring Hg accumulation can be quite a time- and resource-intensive exercise. Resident organisms may be absent due to e.g., seasonal reproductive cycles or in certain situations of strong contamination, while cross-sites comparisons may be hindered by the impossibility of collecting the same organisms at all monitoring points. Bioaccumulation experiments with laboratory-reared organisms, therefore, become attractive to assess the bioaccumulative potential of different chemicals, especially in medium- to large-sized monitoring programs. However, contaminant bioaccumulation even in taxonomically close species can vary because of different ecological traits [12–14] and simple sediment sampling operations can increase the availability of several trace elements in laboratory settings [15]. The use of in situ bioaccumulation tests [16,17] would prevent problems related to sampling and sample handling, but performing in situ exposures in deep waters is subject to the availability of divers and becomes unfeasible for depths above 20 m. Following all these considerations, it is clearly necessary to verify that laboratory tests using individual model species can predict Hg accumulation by indigenous biota with an acceptable degree of accuracy.

In the present study, laboratory-reared specimens of the model benthic organisms *Chironomus riparius* (first-instar larvae) and specimens of *Lumbriculus variegatus* were exposed to sediments collected at various sites in Lake Maggiore (Northwestern Italy), a holo-oligomictic lake still showing signs of legacy Hg pollution from a chlor-alkali plant eventually dismissed in 1998. Assemblages of indigenous chironomids and oligochaetes were also collected and taxonomically identified to gather information about their spatial distribution, and to compare their different feeding strategies. Furthermore, adults of *C. riparius* emerging in the laboratory after larval exposure to field-collected sediments were also analyzed for Hg content to estimate the Hg load that may be exported from contaminated sediments to the surrounding terrestrial environments or to flying insectivores (swifts, swallows, and bats). The main aim of this research was to evaluate if Hg bioaccumulation in indigenous chironomids and oligochaetes (logistically very demanding and subject to the availability of biological material) can be reliably estimated using laboratory bioaccumulation tests with the corresponding model organisms *C. riparius* and *L. variegatus*. A linear relationship between Hg bioaccumulation in model versus indigenous organisms was obtained for chironomids, but not for oligochaetes. Our results also highlighted that sediment preparation for laboratory bioaccumulation tests could modify their total Hg content by up to 2-fold following mixing of sediment layers with different degrees of contamination. From a risk-assessment perspective, laboratory bioaccumulation tests should therefore not be used as the sole tool for decision-making. Analysis of adult specimen of *C. riparius* highlighted a possible role of chironomids in remobilizing Hg to terrestrial ecosystems and deserves further study.

## 2. Materials and Methods

### 2.1. Study Area

Lake Maggiore (Figure 1) is a large and deep lake (212.5 km$^2$; volume 37.5 km$^3$; mean and maximum depths: 177 and 370 m) that has suffered from industrial chemical pollution (including

mercury) for most of the 20th century [18–20]. The lake comprises an elongated main basin oriented along the N-S direction and a sub-basin, the Borromean Gulf, to the west (Figure 1). The Borromean Gulf is directly influenced by water and particulate inputs from River Toce where the main Hg-generating industrial activities were located (Figure 1). The adjacent Lake Mergozzo is connected with the Borromean Gulf via an artificial channel that allows bidirectional water and particle exchange between the two water bodies.

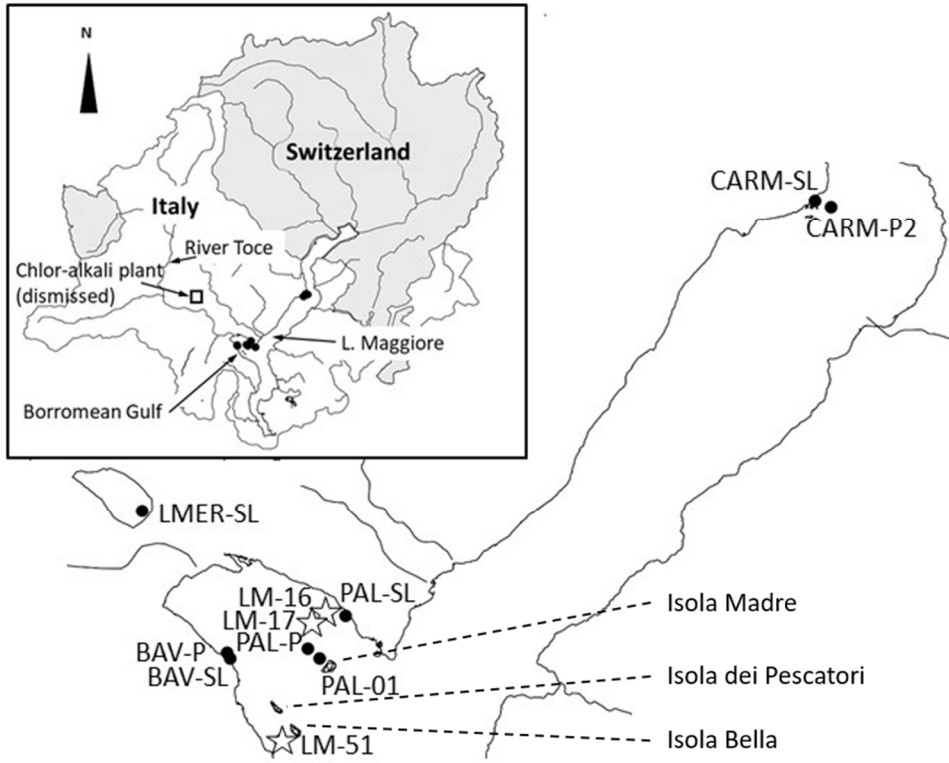

**Figure 1.** Lake Maggiore watershed (Italy/Switzerland): white box = position of the chlor-alkaly plant; black dots = sampling sites for indigenous chironomids, oligochaetes and bottom sediments; white stars = location of sediment cores collected during previous studies [21,22]. See text for details and Table 1 for site codes.

## 2.2. Sampling Methodology

Indigenous organisms and sediments were collected in the Borromean Gulf, in front of the town of Carmine Inferiore (that was outside the area directly affected by industrial Hg polluting sources), and at one site in Lake Mergozzo (Figure 1).

Sediment samples were collected using a 5-L or a 40-L Van Veen grab depending on specific sampling purposes: recovery of indigenous organisms for Hg bioaccumulation, collection of sediments for standardized bioaccumulation tests in the laboratory and recovery of indigenous organisms for taxonomical identification (Figure 2). The whole study was coordinated by the Piedmontese Environmental Protection Agency (ARPA Piedmont, Omegna, Italy) and a synopsis of the actual activities carried out at the sampling sites is provided in Table 1.

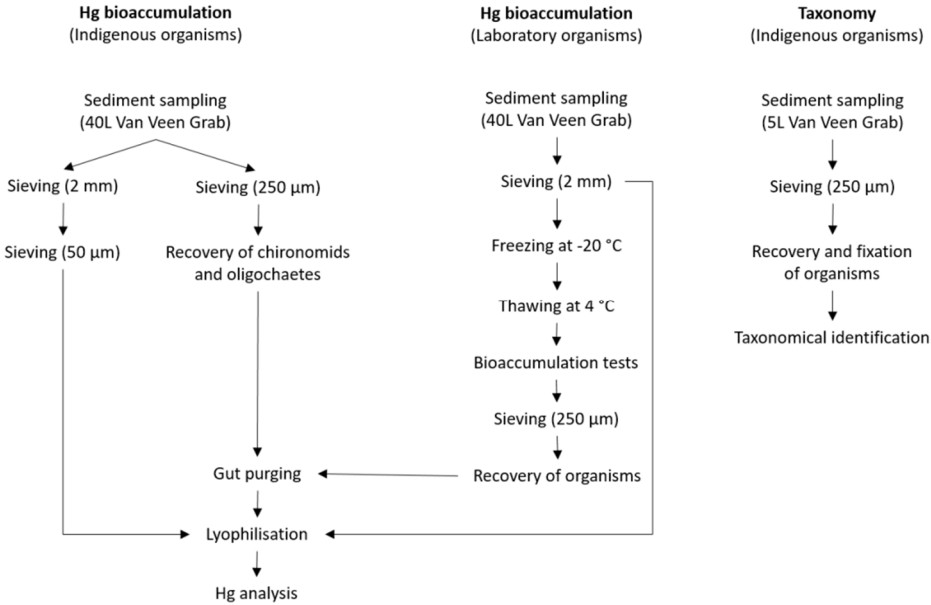

**Figure 2.** Overview of treatment procedures undergone by sediments sampled for different purposes in the course of the present study.

Indigenous chironomids and oligochaetes for Hg analysis were recovered only in Spring 2010 by sieving sediment samples on a 250 μm nylon mesh net directly in the field and then kept in lake water with a thin layer of sediments. Tubicolous chironomids and oligochaetes were sorted in the laboratory, purged in filtered (<1 μm) Lake Maggiore water for 6 h, blot dried and frozen pending lyophilisation and analysis (Figure 2). Aliquots of the same sediments were sieved through 2 mm and 50 μm for determination of total Hg. The use of a 40-L grab sampler was necessary to recover from sediments a sufficient amount of organisms for one Hg analysis of pooled indigenous organisms. The choice of tubicolous chironomids chironomini and oligochaetes tubificidae is related to their feeding habits that make these organisms suitable proxies for studying different habitat conditions. Most tubificids stay with their caudal end protruding out of the top surface sediments (head-down) and serve as a proxy for the strict within-sediment habitat conditions. By contrast, the tube-building chironomids chironomini feed on freshly deposited or on suspended matter [17,23,24] without leaving their tubes and keeping their head exposed to the water (head-up). They serve as a proxy for water-sediment interface habitat conditions.

Sediment samples for Hg bioaccumulation tests in the laboratory were collected in spring 2010, sieved at 2 mm, frozen to kill indigenous organisms and then thawed for preparation of batch exposures in the laboratory (Figure 2and Figure S1). Aliquots of homogenized, sieved sediments were also freeze-dried and analysed for total Hg (Figure 2). As will be discussed later in the text, differences in the mesh-size of sieves were without effect on the total Hg content measured in sediment aliquots.

Organisms for taxonomic identification were recovered in spring and autumn 2010 following the Italian national standardized method [25] according to the Water Framework Directive [26]. The autumn sampling was performed to verify that benthic organisms with a terrestrial adult stage (which include chironomids) were not partially or completely absent from sediments collected in spring due to the emergence of their images. The methodology provided for taking samples along transects at different depths representing the littoral, sublittoral and deep zones of a lake. In the present study, littoral stations (less than 25 m deep) were excluded because tubicolous chironomids and oligochaetes could be more easily found at higher depths. Three transects were, therefore, considered (one in front of Pallanza municipality, one in front of Baveno, and one in front of Carmine Inferiore), each one including a sub-littoral (SL) and a profundal (P) station (Table 1 and Figure 1). Their distribution within the Borromean Gulf was based on previous knowledge of Hg spatial contamination to include sites

with both high and low total Hg contents (Figure 1, Tables 1 and 2) [27,28]. All samples of indigenous organisms were first washed with lake water using a 250 µm mesh aperture net to allow the collection of even the smallest species. A 10% formalin solution, neutral buffered, was then used to fix samples and inhibit autolysis and putrefaction. Sieving, sorting (under a stereo microscope Leica 80×) and taxonomic identification of organisms (through a Zeiss microscope) were performed in the laboratory. The taxonomic identification based on morphological and phenotypic characteristics was carried out for oligochaetes [29], chironomids [30,31] and the remaining groups of macroinvertebrates [32]. Species level of identification was generally preferred, but sometimes a higher taxonomic rank (genus or family) has been employed because of the presence of unidentifiable juveniles.

**Table 1.** Information on sampling sites and activities performed in lakes Maggiore and Mergozzo for the present study. Cores LM16, LM16 and LM51 were collected during previous studies [21,22] and used for data interpretation and comparison purposes. OC (%): organic carbon (in %); Tax: taxonomical identification (Yes/No); $[Hg]_{ind}$: total Hg measurements in indigenous organisms (Y); $[Hg]_{Lab}$: bioaccumulation tests with Ch (*Chironomus riparius*) and Lv (*Lumbriculus variegatus*) (Y/N); —: not analyzed or not performed.

| Sampling Site | Code | Coordinates | | Depth | OC (%) | Tax | $[Hg]_{ind}$ | $[Hg]_{Lab\ Ch}$ | $Hg_{Lab\ Lv}$ |
|---|---|---|---|---|---|---|---|---|---|
| Pallanza sublittoral | PAL-SL | 464,649.7 | 5,085,949.2 | 28 | 2.04 | Y | Y | Y | Y |
| Pallanza deep | PAL-P | 463,631.9 | 5,085,064.0 | 115 | 1.10 | Y | Y | Y | Y |
| Baveno sublittoral | BAV-SL | 461,540.2 | 5,084,798.2 | 28 | 1.44 | Y | Y | Y | Y |
| Baveno deep | BAV-P | 461,457.8 | 5,084,955.1 | 54 | 1.12 | Y | Y | Y | Y |
| Carmine sublittoral | CARM-SL | 477,292.6 | 5,097,150.2 | 29 | 1.32 | Y | Y | Y | Y |
| Carmine deep | CARM-P2 | 477,725.2 | 5,096,980.3 | 113 | 1.30 | Y | Y | N | Y |
| Borromean Gulf | PAL-01 | 463,950.3 | 5,084,800.0 | 30 | 1.14 | N | Y | Y | N |
| L. Mergozzo sublittoral | LMER-SL | 459,174.7 | 5,088,792.7 | 25 | 2.97 | N | Y | N | Y |
| Borromean Gulf east side | LM16 | 464,140.3 | 5,086,003.8 | 118 | — | — | — | — | — |
| Borromean Gulf east side | LM17 | 463,707.9 | 5,085,728.5 | 123 | — | — | — | — | — |
| Borromean Gulf west side | LM51 | 462,892.1 | 5,082,585.3 | 59 | — | — | — | — | — |

**Table 2.** Total mercury concentrations (mg kg$^{-1}$ dry weight) in sediments, and indigenous and laboratory-exposed organisms. Field$_{Chiro/Oligo}$: sediment aliquots for indigenous Hg analysis; Lab$_{Ch}$ and Lab$_{Oli}$: sediment aliquots used for bioaccumulation tests on *Chironomus riparius* and *Lumbriculus variegatus*; Ch$_{Ind}$ and Oli$_{Ind}$: indigenous chironomids and oligochaetes; Ch$_{Lab}$ and Lv$_{Lab}$: *C. riparius* and *L. variegatus* exposed to tests; BSAF: biota-sediment accumulation factor; CNT: control exposures; n.a.: not applicable; n.m.: not measured. Concentrations with $ were not measured and values from other aliquots were considered for calculation and comparison purposes.

| Code | Sediments | | | Chironomids | | Oligochaetes | | BSAF | | | |
|---|---|---|---|---|---|---|---|---|---|---|---|
| | Field$_{Chiro/Oligo}$ | Lab$_{Ch}$ | Lab$_{Oli}$ | Ch$_{Ind}$ | Ch$_{Lab}$ | Oli$_{ind}$ | Lv$_{Lab}$ | Ch$_{Ind}$ | Ch$_{Lab}$ | Oli$_{Ind}$ | Lv$_{Lab}$ |
| PAL-SL | 4.441 ± 0.085 | 8.876 ± 0.279 | 8.51 ± 0.384 | 0.396 | 2.020 | 0.856 | 3.588 ± 1.078 | 0.09 | 0.23 | 0.19 | 0.42 |
| PAL-P | 0.233 ± 0.020 | 0.353 ± 0.015 | 0.334 ± 0.014 | 0.125 | 0.214 | 0.278 | 0.169 ± 0.012 | 0.54 | 0.61 | 1.19 | 0.50 |
| BAV-SL | 0.137 ± 0.003 | 0.181 ± 0.026 | 0.132 ± 0.006 | 0.302 | 0.249 | 0.134 | 0.445 ± 0.391 | 2.20 | 1.38 | 0.98 | 3.37 |
| BAV-P | 0.175 ± 0.012 | 0.214 ± 0.030 | 0.186 ± 0.012 | 0.180 | 0.302 | 0.247 | 0.145 ± 0.013 | 1.03 | 1.41 | 1.41 | 0.80 |
| CARM-SL | 0.122 ± 0.001 | 0.098 ± 0.003 | 0.098 ± 0.001 $ | 0.105 | 0.152 | 0.512 | 0.220 ± 0.007 | 0.86 | 1.55 | 4.20 | 2.24 |
| CARM-P2 | 0.258 ± 0.006 | n.m. | 0.258 ± 0.006 $ | n.a. | n.a | 0.270 | 0.153 ± 0.006 | n.a. | n.a | 1.04 | 0.59 |
| PAL-01 | 6.328 ± 0.061 | 4.998 ± 0.053 | 4.998 ± 0.053 $ | 0.875 | 1.47 | 1.752 | n.a. | 0.14 | 0.29 | 0.28 | n.a |
| LMER-SL | 1.479 ± 0.019 | n.m. | 1.479 ± 0.019 $ | 0.964 | n.a. | 0.395 | 0.241 ± 0.034 | 0.65 | n.a | 0.27 | 0.16 |
| CNT | n.a. | 0.018 ± 0.001 | n.m. | n.a. | 0.279 | n.a. | 0.220 ± 0.098 | n.a. | n.a. | n.a. | n.a. |

### 2.3. Test with Chironomus riparius

*Chironomus riparius* Meigen 1804 (Diptera, Chironomidae) adults were initially obtained from a culture held at the laboratory ECT Oekotoxikologie GmbH., Florsheim Main (Germany). Chironomids were reared in 40 L aquaria containing a 4 cm layer of sand (<250 μm) and 8–10 cm of aerated tap water. The culture was kept at 20 ± 2 °C under a daily photoperiod (16:8 h light:dark; 800 lux). The overlying water was renewed every other day and organisms were fed weekly with Tetramin® fish food (Tetrawerke, Melle, Germany) at 1 mg per organism. Bioaccumulation tests were carried out following the main indications of the Organisation for Economic Co-operation and Development (OECD) guidelines 218 and 219 [33,34] and lasted 10 days.

Sediments samples were frozen at −20 °C to eliminate indigenous macrofauna and transferred at 4 °C for one week before the start of the bioaccumulation experiment. Three days before the addition of the animals, 250 mL glass beakers were filled with 50 mL of wet weight sediment (combed through 2 mm sieve to eliminate debris and coarse material that could have perturbed the well-being of test organisms) and 200 mL of filtered (1 μm), aerated Lake Maggiore water gently added to the sediment. About 15 L of Lake Maggiore water were collected in the middle of the basin in the pelagic area of the lake on a single day during the sampling period, filtered directly on site and stored at 4 °C in the dark until use. The typical composition of Lake Maggiore water in the topmost 25 m of the water column in March 2010 (regular monitoring of CNR-IRSA, Verbania-Pallanza offices) was as follows: Ca = 22.6 mg L$^{-1}$, Mg = 3.68 mg L$^{-1}$, Na = 3.06 mg L$^{-1}$, K = 1.51 mg L$^{-1}$, Cl = 2.96 mg L$^{-1}$, SO$_4$ = 29.2 mg L$^{-1}$, N-NO$_3$ = 855 μg L$^{-1}$, total organic carbon = 0.7 mg L$^{-1}$, hardness = 72 mg L$^{-1}$ as CaCO$_3$, conductivity = 150 μS cm$^{-1}$, alkalinity = 0.829 meq L$^{-1}$, pH = 7.53. To allow sediments and water to equilibrate, the beakers were covered with a plastic Petri dish and kept in the dark at 20 ± 1 °C and without aeration for three days (Figure S1).

Five replicate beakers were prepared for each site along with a laboratory control consisting of artificial clean sediment prepared according to OECD 218/219 [33,34]. At the start of the test, the overlying water of each beaker was gently aerated for 2 h and 15 first instar larvae chosen at random were transferred to all replicate and control beakers. Tests were performed under artificial light (800 lux) with a 16:8 h light:dark photoperiod. The animals were fed three times a week with a Tetramin® suspension at 0.4 mg per organisms for the first 10 days of exposure and at 0.7 mg per organisms for the rest of the test. The water lost through evaporation was replaced with spring water. Temperature, pH, conductivity and dissolved oxygen were measured in all the beakers at the beginning and at the end of the tests. Organisms in two replicate beakers were manually recovered after 10 days, weighted, kept in lake water without feeding for 8 h to purge their guts, pooled and freeze-dried for Hg analysis. In the other three replicates, covered with nets, exposure continued until 25–28 days to collect emerging adults for Hg analysis. Each replicate was gently aerated with a flow-rate of 1–2 bubbles per second.

### 2.4. Test with Lumbriculus variegatus

The laboratory culture of *Lumbriculus variegatus* (Müller, 1774) originated from the University of Joensuu (Department of Biology) Finland. The oligochaetes were reared in several 1 L-beakers containing 200 mL of clean sand and 700 mL of reconstituted water prepared according to United States Environmental Protection Agency (USEPA) methodology [35] by dissolving the following salts in deionized water: 0.05 g L$^{-1}$ CaSO$_4$, 0.05 g L$^{-1}$ CaCl$_2$, 0.03 g L$^{-1}$ MgSO$_4$, 0.096 g L$^{-1}$ NaHCO$_3$, 0.004 g L$^{-1}$ KCl. The reconstituted water had the following characteristics: hardness 106 ± 4 mg L$^{-1}$ as CaCO$_3$, alkalinity 1.2 ± 0.1 meq L$^{-1}$, conductivity 318 ± 34 μS cm$^{-1}$, and pH 8.07 ± 0.42. The beakers were held in a temperature-controlled water bath at 20 ± 1 °C under a 16:8 h light:dark photoperiod at about 150 lux. The worms were fed with approximately 100 mg ground fish food (Tetramin®, TetraWerke, Melle, Germany) twice a week. Overlying water in the beakers was continuously aerated with a Pasteur pipette and it was manually renewed at the same time as feeding. Water renewal was necessary to prevent bacterial and fungal growth as well as excess levels of ammonia due to the accumulation of excess food and excretory products.

A whole-sediment bioaccumulation test was conducted based on standard methods [35–37] with minor modifications. The photoperiod, temperature and aeration were the same used for the laboratory culture. Exposures were conducted in 3-L glass vessels (22 cm i.d., 9 cm ht) containing 0.6 L of whole sediment and 2.4 L of reconstituted water that was continuously aerated (1 bubble per second). The vessels were covered to limit the loss of overlying water. Before worm addition, sediments in test vessels were conditioned for 7 days at test temperature to stabilize ammonia concentration in the overlying water. The overlying water lost via evaporation or collected for overlying water quality analysis was filled up with deionised water. Oligochaetes were not fed during the sediment exposures.

Control vessels were prepared as described above using sand and USEPA reconstituted water. The worms in the control vessels were fed three times a week with approximately 150 mg ground fish food (Tetramin®). The difference in the amount of added food with respect to the laboratory culture (see above) account for the different dimensions of the vessels and the different numbers of worms in the vessels. Half volume of the overlying water of the control vessels was manually renewed at the same time as feeding for the reasons already explained for the culture of *L. variegatus*.

On the day before the start of the exposures (day-1), mixed-age oligochaetes were manually removed from the culture, rinsed to remove most of the debris and placed into 1 L beakers containing aerated USEPA reconstituted water. On day 0, the oligochaetes were combined into a glass pan and rinsed to break up the masses of oligochaetes and remove any remaining debris. Weighted aliquots of oligochaetes (mean percent dry weight $14 \pm 0.4$ mg), which were fixed according to ratio of total organic carbon in sediment (Table 1) to organism dry weight $\geq 50$ [35], were transferred to the different exposure units (see Table S1 for the number of replicates used for different sediments). The weight of the aliquots of worms were carried out by gently filtering them ($-0,8$ kPa, 60 s) on a pre-weighed polycarbonate filter (Nuclepore 3.0 μm, 47 mm i.d.) in order to remove the excess water [38].

Operation of the exposure system, dissolved oxygen, temperature and behaviour of oligochaetes were checked daily. Conductivity, pH, total hardness, alkalinity and total ammonia of the overlying water were measured at the beginning of the exposures (before adding the organisms) and then three times a week. When the ammonia concentration was above 1 mg N L$^{-1}$ half volume of the overlying water was manually renewed.

At the end of the exposure time (day 28), oligochaetes were recovered by sieving through a 250 μm-mesh sieve, and then collecting the contents of each sieve in a glass pan. The oligochaetes were sorted from detritus and kept for 6 h into 1 L beakers containing aerated reconstituted fresh water for gut purging [35,39]. The worms were then rinsed to remove any remaining debris, weighted after having removed the excess of water by filtration as reported above, and frozen at $-20$ °C pending lyophilisation and analysis.

*2.5. Mercury and Organic Carbon Analyses*

All Hg analyses were performed by atomic absorption spectroscopy using an automated Hg analyser (AMA254, FKV, Sorisole, Bergamo, Italy) and all sediment samples were analysed in triplicate (50–100 mg for each replicate). For laboratory tests with *C. riparius*, sediment samples were collected from each of the three experimental replicates incubated for 25–28 days to avoid losing or damaging the chironomids specimen to be used for Hg bioaccumulation at day 10. For the tests with *L. variegatus*, sediment samples were collected from each exposure aquarium. Blanks were systematically run between samples and, if necessary, repeated until Hg levels fell to 0.002 absorbance unit (approx. 0.06 ng Hg) as per instrument instructions.

Mercury content in indigenous chironomids and oligochaetes was determined on one pool of individuals collected as described in Section 2.2. For laboratory organisms, specimens of *C. riparius* from two exposure replicates were pooled for Hg analysis, while the exposed individuals of *L. variegatus* were assayed independently for each exposure replicate.

Accuracy was verified with certified reference materials GBW07305 (stream sediment) from the National Standard Centre of China ($0.1 \pm 0.02$ mg kg$^{-1}$) and with internal reference materials

S7 (stream sediment) used in the Analytical Quality Control and Assessment (AQUACON) project ($1.72 \pm 0.03$ mg kg$^{-1}$) and T13 (mussel tissue, $0.188 \pm 0.02$ mg kg$^{-1}$). Average recoveries (arithmetic means $\pm$ 1 standard deviations; n = 6) were $0.099 \pm 0.004$ mg kg$^{-1}$ for GBW07305, $1.95 \pm 0.036$ mg kg$^{-1}$ for S7, and $0.195 \pm 0.003$ for T13. Typical precision for replicate sediment measurements was 5% or better. Replicate measurements for biological material were possible only for laboratory-exposed *L. variegatus* (see data in Table 2 and Section 3.3). Variability in control organisms was 44%. It is important to note that the pools of *L. variegatus* for Hg analysis were not sub-samples of the same batch, but originated from different exposure batches.

The organic carbon content in sediment samples (aliquots used for laboratory Hg bioaccumulation) was determined by oxidation with potassium dichromate using the Walkley–Black method [40]. Briefly, an oxidation mixture was prepared using 10 mL of a 1 N$K_2Cr_2O_7$ solution and 20 mL of concentrated sulphuric acid (96%). The reaction was stopped after 30 min by adding 200 mL of deionized water and the resulting solution was titrated using 1,10-phenanthroline.

### 2.6. Data Treatment and Statistics

Variables were tested for normality using the Shapiro–Wilk test. Due to the presence of non-normal variables (see Sections 3.2 and 3.3) comparison of the central tendencies were performed using the rank sum test based on median values. Correlations were determined using the Spearman correlation coefficient, while linear regression were obtained using a robust MM estimator (library MASS in R version 3.6.0). Robust linear regression are less influenced by the presence of outliers of non-normally distributed errors [41] and were preferred to the standard least-squares method. Note that only paired data were used for all statistical analyses.

## 3. Results

### 3.1. Mercury Levels in Sediments

Total Hg concentrations in sediments ranged from values only slightly above the background level $0.044 \pm 0.026$ mg kg$^{-1}$ estimated by Vignati and Guilizzoni [42] for the Borromean Gulf to situations indicating the presence of important anthropogenic contamination (Table 2). An earlier study by Viel and Damiani [43] collected 126 samples of surface sediments in the whole basin of Lake Maggiore and statistically estimated a background value of 0.03 mg kg$^{-1}$ d.w. for total Hg concentration. The same study reported also reported a value of 0.2 mg kg$^{-1}$ d.w. total Hg for turbiditic sediments.

Different Hg contents in sediments collected from sites located along the same transect are most likely related to the water-depositional dynamics. In the Borromean Gulf, the particulate material carried by the River Toce tends to sink between the Isola Madre to the northeast and the Isola Bella and Isola dei Pescatori to the south-west (Figure 1) following the steepest depth gradient. On the other hand, the area between the Isola Madre and Pallanza shows a decreasing depth because of the emergence of Isola Madre from the lakebed. This low-depth area is the remains of a barrier that rises from a submerged valley floor at the former confluence of the past glacial valleys formed by therivers Toce and Ticino. Given these morphological characteristics of the lake bottom, part of the solid load may be deposited in the littoral zone and contribute to the observed Hg enrichment at the Pallanza sublittoral (PAL-SL) compared with the Pallanza deep (PAL-P). Furthermore, the predominant south winds create anticlockwise currents in the Borromean Gulf [44]. These currents can transport the fine material carried by River Toce to the northern part of the Borromean Gulf and then outside the bay towards the southern part of the lake. Finally, Hg contamination inside the Borromean Gulf seems to be heterogeneous both horizontally and vertically; which could explain the different Hg levels at PAL-P versus PAL$-01$ despite their spatial proximity [27]. The spatial heterogeneity in Hg sediment content is also the most likely cause of the variability in total Hg concentrations between sediments aliquots used to isolate indigenous organisms (Field$_{chiro/oligo}$ in Table 2) and those used for laboratory bioaccumulation experiments (Lab$_{Ch}$ and Lab$_{Oli}$ in Table 2).

Lake Maggiore has suffered a long history of Hg pollution [18,20], although the Hg contamination levels in sediments have been decreasing since the 1970s [18]. The calculated average sedimentation rate in the Borromean Gulf is $0.46 \pm 0.05$ cm $yr^{-1}$ (n = 5) over the period 1989–2014 [45]. Similar sedimentation rates (about 0.5 cm $yr^{-1}$) have been reported in the main basin of Lake Maggiore, albeit the amount of available information is less exhaustive in this case [46]. Based on these figures, the thickness of (relatively) clean sediments covering the most contaminated layers (dating back to about 1970s and 1950s; [18]) should be about 20–25 cm. However, sediment slumps of other dynamic phenomena can alter the sedimentation pattern and increase the patchiness in the spatial distribution of contaminated sediments. The sampling devices used for different purposes of the present study (see Section 2.2 and Figure 2) likely penetrated at different depths into sediments and the resulting homogenised samples used for Hg analysis may well have integrated layers with variable levels of contamination. Most importantly, ARPA Piedmont, the coordinator of the whole research programme (see Section 2.2) verified that total Hg in sediment aliquots sieved at 2 mm or at 50 μm agreed to 10% or better (data not shown). This observation agrees with the general knowledge of grain size distribution in the Borromean Gulf where the presence of sandy coarse sediments is limited to the areas closed to the mouth of River Toce, while silts and clay predominate in the rest of the Borromean Gulf [47], including the sampling sites considered in the present study. Furthermore, it is easily shown that freezing of sediments (sieved at 2 mm) to kill indigenous organisms before laboratory bioaccumulation experiments is very unlikely to increase the total Hg sediment content by more than 1% (see the Supporting Information for detailed theoretical calculations). In conclusion, differences in sediment sieving linked to different methodological constraints and standardized procedures were not the cause of the observed differences in Hg content.

The mercury content of sediments collected from Lake Mergozzo (Table 2) confirmed that Hg contamination also reached this water body via either the connecting channel with Lake Maggiore (see Section 2.1) or atmospheric deposition. Average mercury levels in the topmost 10 cm of a core collected from Lake Mergozzo in another study carried out in 2011 were $0.78 \pm 0.14$ mg $kg^{-1}$ d.w. (arithmetic mean ± 1 standard deviation, range 0.62–0.98 mg $kg^{-1}$; our unpublished data). Mercury levels at the bottom of the same core were around 0.1 mg $kg^{-1}$, suggesting the presence of persistent Hg contamination also in this water body.

### 3.2. Bioaccumulation in Indigenous Organisms

Mercury levels were usually higher in strictly sediment feeding (detritivorous) oligochaetes than in chironomids (Table 2). However, median values (0.395 vs. 0.302 mg $kg^{-1}$, n = 7) were not statistically different ($p$ = 0.602, Rank Sum Test). Results for site Carmine deep (CARM-P2) were excluded from the statistics because only oligochaetes could be collected for Hg analysis at this site. Mercury concentrations in indigenous chironomids and oligochaetes were not correlated with total Hg content in sediments (Spearman rho = 0.75, $p$ = 0.066, n = 7 for chironomids; rho = 0.619, $p$ = 0.105, n = 8 for oligochaetes) and between themselves (Spearman rho = 0.321; $p$ > 0.5, n = 7 considering only sites where both groups could be sampled and analysed—Table 2). The largest potential for Hg transfer from sediment to benthic organisms was observed at stations Baveno sublittoral (BAV-SL), Baveno, Pallanza and Carmine deep (BAV-P, PAL-P, CARM-P2) and Carmine sublittoral (CARM-SL) that showed low to moderate Hg enrichment in sediments. At these sites, biota-sediment accumulation factors (BSAF), defined as the ratio between Hg concentrations measured in organisms and in sediments, ranged from about 0.5 to 4 (Table 2), while BSAF for organisms collected at the more contaminated sites PAL-SL and PAL-01 did not exceed 0.4. The situation for chironomids from Lake Mergozzo may be peculiar and deserves further study.

Differences in Hg accumulation among indigenous chironomids and oligochaetes at a given site reflect the combined outcome of organisms' ecology and patterns of Hg contamination in surface sediments. Results obtained from sediment cores LM16, LM17 and LM 51 collected in the Borromean Gulf in 2011 (Figure 1 and Table 1) provide a good example. The three cores have similar sedimentation

rates of about 0.7 cm year$^{-1}$ (Table S2), but different Hg contamination profiles over the period 2005–2011 covering three separated sediment layers. Hg levels in core LM16 were homogeneous (0.247 ± 0.037 mg kg$^{-1}$ for the three layers). Cores LM17 and LM51 had similar Hg levels in the top 3 cm of sediments (2007–2011), which overlaid more contaminated sediment layers deposited in 2005–2007 (Table S2). Vertical variability of Hg levels in (sub)surface sediments can be particularly important in explaining bioaccumulation differences between tube-building head-up chironomids feeding on the surrounding sediment surface or intercepting drifting food and tube-building head-down oligochaetes that burrow deeper acting as bioturbators. On the other hand, chironomids can feed on suspended matter freshly deposited around their tubes [17]. Such additional route of exposure may be of variable importance depending on the actual sedimentation rates in the different areas.

*3.3. Bioaccumulation in Laboratory-Reared Organisms*

In tests with *C. riparius*, exposure temperatures, pH values and oxygen and ammonia concentrations remained within acceptable values and within 10% of the initial values for all exposure conditions. A similar situation was observed for tests with *L. variegatus* with the exception of N-NH$_4$ which tended to decrease over time (data not shown). As specified in Section 2.4, overlying water in the tests with *L. variegatus* was renewed as needed when N-NH$_4$ levels became higher that 1 mg L$^{-1}$. The observed differences in Hg bioaccumulation should therefore reflect variations in Hg bioavailability from sediments collected at the various locations independently of uncontrolled deviations in master hydrochemical parameters in certain exposure beakers. After 10 day of exposure, chironomids survival in control organism was 100% and above 80% in all sediment except CARM-SL (66%). Emergence at day 28 was 100% for control organisms and above 70% for all sediments. The absence of mortality in controls guarantees good test conditions and the survival of organisms exposed to field sediments was high enough to recover sufficient material for Hg analyses.

Direct comparison of accumulated Hg concentrations in *C. riparius* and *L. variegatus* was possible for sites PAL-SL, PAL-P, BAV-SL, BAV-P and CARM-SL. Median Hg levels were similar for the two organisms (0.249 vs. 0.220 mg kg$^{-1}$, n = 5). On the other hand, no correlation existed between Hg accumulation in *C. riparius* versus *L. variegatus* (Spearman rho = 0.68; $p > 0.5$, n = 5 considering only sediments that were tested with both organisms) or with the corresponding Hg concentrations measured in test sediments ($p = 0.058$ for tests with *C. riparius*; $p = 0.5$ for tests with *L. variegatus*). In laboratory settings, sediments were homogenized before use and organisms' ecology is expected to be less important in determining bioaccumulation differences than under real field conditions.

The relative standard deviation among replicate Hg bioaccumulation analyses in the experiments with *L. variegatus* (n = 2–9; see Table S1) were within 15% except for PAL-SL (30%) and BAV-SL (80%). For both samples, only two replicate measurements could be performed due to availability of sediments and the variability among measurements was calculated using the formula:

[Absolute value (Measurement 1 − Measurement 2)]/square root of 2

The formation of two distinct sediment layers, a light brown upper layer about 0.5 cm thick covering a thicker layer of black sediment, was observed in the aquaria used for *L. variegatus* exposures. Analyses of Hg concentration in the two layers of selected sediment samples usually agreed to within 10% (Table S3).

Mercury content in control organisms was between 0.2 and 0.28 mg kg$^{-1}$; comparable to levels measured in specimen exposed to lake sediments excluding the most contaminated sediment samples (i.e., PAL-SL and, for *C. riparius* only, PAL-01). The use of filtered Lake Maggiore water (tests with *C. riparius*) vs. synthetic water (tests with *L. variegatus*) does not seem to have introduced any systematic bias in Hg accumulation during laboratory tests. An important contribution of the aqueous phase to Hg accumulation should have resulted in a systematically higher Hg burden of *C. riparius* compared with *L. variegatus*; which was not observed (Table 2). Although the Hg content in filtered Lake Maggiore

waters was not determined, this observation suggests that most of the Hg burden in test organisms was linked to dietary uptake. The Tetramin® food used to feed *C. riparius* and control *L. variegatus* during the tests contained $0.051 \pm 0.001$ mg kg$^{-1}$ d.w. Hg (n = 3). In the case of chironomids, 0.4 g Tetramin® per larva per day were administered over the 10-day test [33,34]. The pooled control sample that was analysed for Hg consisted of 30 specimens which, taken collectively, were fed 120 mg Tetramin®. This amount of food contained an absolute quantity of 6.12 ng Hg. Assuming that all the added food and associated Hg would have been ingested and assimilated by chironomids, one would expect an Hg concentration of about 0.402 mg kg$^{-1}$ in control organisms. Complete ingestion and assimilation are, however, unlikely and these simple calculations do not take into account the dynamic nature of contaminant accumulation in invertebrates [48,49]. Overall, the addition of Tetramin® could explain the Hg accumulation observed in control chironomids, considering that *C. riparius* has been shown to feed preferentially on added food during laboratory tests [50]. However, it remains an open question whether Hg levels in *C. riparius* exposed to field sediments were determined solely by the addition of Tetramin®.

Organisms exposed to field-collected sediment likely had other carbon sources available besides Tetramin® (see the OC levels in Table 1) and larvae exposed to some field-collected sediments eventually accumulated less Hg than control larvae. Indeed, the addition of food can alter contaminant bioavailability in different ways depending on, for example, the contaminant getting adsorbed on the added food, which increases contaminant bioavailability, or not [50]. Except in control sediments, Hg concentration in Tetramin® was lower than Hg concentration in sediments (Table 2) implying that Hg bioaccumulation caused by sediment ingestion could have been partially diluted by the ingestion of cleaner Tetramin® particles. It is also important to remember that specimen of *L. variegatus* exposed to lake sediments were not fed during the bioaccumulation tests. In this case, the accumulated Hg therefore reflect uptake from the element pool present in the sediment themselves.

The differences in food addition between the two tests considered the provisions made by the corresponding norms. The OECD norms for *C. riparius* [33,34] were developed for ecotoxicity testing of artificial spiked sediments in contrast to the USEPA, OECD and American Society for Testing and Materials (ASTM) norms for *L. variegatus* [35–37] that specifically focus on bioaccumulation. These differences may influence the comparison between the two organisms with regard to accumulated Hg in laboratory settings. On the other hand, these differences do not impact the main purpose of the present work, that consists in comparing Hg accumulation in indigenous vs. laboratory reared organisms. In theory, a preferential Hg accumulation from food in laboratory-reared *C. riparius* should bring about a fixed extra-Hg accumulation compared with indigenous organisms. As explained in the previous paragraph, Hg accumulation from added food does not seem to have been the sole source of Hg during the tests with *C. riparius*. Further arguments in support of a moderate effect of added food on Hg accumulation are presented in Section 3.4.

After 10 days of exposure, larvae of *C. riparius* exhibited mortality >20% only for CARM-SL site. Similarly, larvae grown in CARM-SL sediments had the lowest average fresh weight (2.1 vs. 5.1 mg ind$^{-1}$ for control larvae). Average weight at the other sites ranged from 3.3 to 4.2 mg ind$^{-1}$. Appreciating sediment ecotoxicity goes beyond the scope of the present paper, but two aspects deserve a brief mention in this respect. The actual role of accumulated Hg in eliciting the observed biological effects may be assessed by comparing measured Hg concentrations with the corresponding critical body residue (CBR) for Hg or MMHg. McElroy et al. [51] determined average CBRs for MMHg of 10.5 mg kg$^{-1}$ wet tissue (w.w.) for lethality and 0.57 mg kg$^{-1}$ w.w. for reproductive effects in fish and invertebrates. Naimo et al. [52] estimated CBR of 0.046 mg kg$^{-1}$ w.w. as the no observed effect concentration (NOEC) for growth of hexagenid mayflies larvae. Assuming an average moisture content of 79% for chironomids [11], accumulation of total Hg in our pools of *C. riparius* ranged from 0.032 (CARM-SL) to 0.45 (PAL-SL) mg kg$^{-1}$ w.w. The growth impairment observed at CARM-SL is therefore unlikely to originate from Hg accumulation. The CBRs for MMHg may be approached or exceeded at some sites, but their applicability to chironomids deserves further studies. Alternatively, Lake

Maggiore sediments contain, at least for some locations, other contaminants such as legacy DDT [18,19] that could contribute to the observed effects on growth.

As observed for the indigenous organisms, the largest BSAF were obtained for the relatively clean sites BAV-SL, BAV-P and CARM-SL (Figure 1), but no clear pattern between sediment contamination and BSAF could be established. Furthermore, Hg concentrations in sediments sometimes varied between aliquots used in laboratory tests and aliquots collected for direct analysis and used for comparison with indigenous organisms by more than 20% (Table 2). The possible causes for these differences have been discussed in Section 3.1.

The test with *C. riparius* also allowed the recovery and analysis for Hg content of adult images emerging from the sediments. Mercury content in images ranged from 0.116 mg kg$^{-1}$ d.w. for organisms exposed to CARM-SL sediments to 0.571 mg kg$^{-1}$ for those exposed to PAL-01 sediments. Control images contained 0.097 mg kg$^{-1}$ (Table S4). Tremblay et al. [53] measured total Hg contents between 0.153 and 0.230 mg kg$^{-1}$ d.w. in emerging adults of *C. riparius* from reservoirs (total Hg in sediments: 0.050–0.230 mg kg$^{-1}$ d.w.) and similar Hg levels (0.197–0.225 mg kg$^{-1}$) in other dipterans belonging to the chironomini tribe in the absence of a more detailed taxonomic identification. Levels of Hg in chironomini from a control natural lake were 0.057 mg kg$^{-1}$. The percentage of Hg content remaining in the images (as compared with measurements in larvae after 10 days' exposure) was above 50% for the sites with low Hg levels in sediments and decreased to 40% and 24% for PAL-01 and PAL-SL, respectively. In contrast with these findings, Chételat et al. [54] also showed that MMHg concentrations in chironomids increased from larvae to pupae to adults. Further study on the importance of emerging insects in the Hg cycle in some areas of Lake Maggiore is therefore granted.

### 3.4. Predicting Hg Bioaccumulation in Indigenous Organisms from Laboratory Bioaccumulation Experiments

Mercury concentrations in indigenous and laboratory organisms showed a good correlation in the case of chironomids (Spearman rho = 0.88, $p$ = 0.033, n = 6), but no correlation in the case of oligochaetes (Spearman rho = 0.43, $p$ = 0.35, n = 7). The resulting robust linear regression to predict Hg levels in indigenous chironomids based on results from standardized tests with *C. riparius* was:

$$[Hg]_{ChInd} = (0.566 \pm 0.047) \cdot [Hg]_{ChLab} + (0.045 \pm 0.049) \tag{1}$$

where $[Hg]_{ChInd}$ is the Hg concentrations estimated in indigenous chironomids using regression (1), $[Hg]_{ChLab}$ is the accumulated mercury concentrations measured in *C. riparius* after 10-days exposure to field sediments in the laboratory, and all values are in mg kg$^{-1}$ d.w. According to the regression's slope, laboratory chironomids seemed to accumulate about twice as much Hg as indigenous organisms. It is noteworthy that Hg levels in 'Lab$_{Ch}$ sediments' (see Table 2) were also correlated with those in 'Field' sediments (Spearman rho = 0.94, $p$ = 0.017, n = 6) and that the slope of the corresponding linear regression was:

$$[Hg]_{SedimentField} = (0.493 \pm 0.002) \cdot [Hg]_{SedimentChLab} + (0.062 \pm 0.007) \tag{2}$$

where $[Hg]_{SedimentField}$ is the mercury concentration estimated in field sediments from where indigenous organisms were recovered for Hg analysis using regression (2), $[Hg]_{SedimentChLab}$ is the Hg concentration measured in sediments used to expose *C. riparius* in the laboratory (Table 2), and all values are in mg kg$^{-1}$ d.w. Regression (2) shows that Hg concentrations in sediments used for laboratory tests were, on average, twice those measured in sediment aliquots used to isolate indigenous organisms. With two exceptions (see below), differences between measured and predicted Hg concentrations were less than 40% for indigenous chironomids and less than 10% for the 'Field' sediments (Figure 3). Regressions (1) and (2), therefore, suggest that the doubling of Hg accumulation in laboratory sediments used during the bioaccumulation tests with *C. riparius* was responsible for the doubling of Hg accumulation in laboratory organisms compared with indigenous chironomid assemblages. Under such hypothesis, food addition during the laboratory tests with *C. riparius* would be of minor importance in determining the measured

Hg body burdens. The possible factors causing the observed differences in Hg concentrations among sediment aliquots have been discussed in Sections 3.1 and 3.2. On the other hand, regression (1) overestimated Hg levels in indigenous chironomids from site PAL-SL by about 3-fold and regression (2) underestimated Hg concentration in 'Field' sediments from site PAL-01 by 2.5-fold.

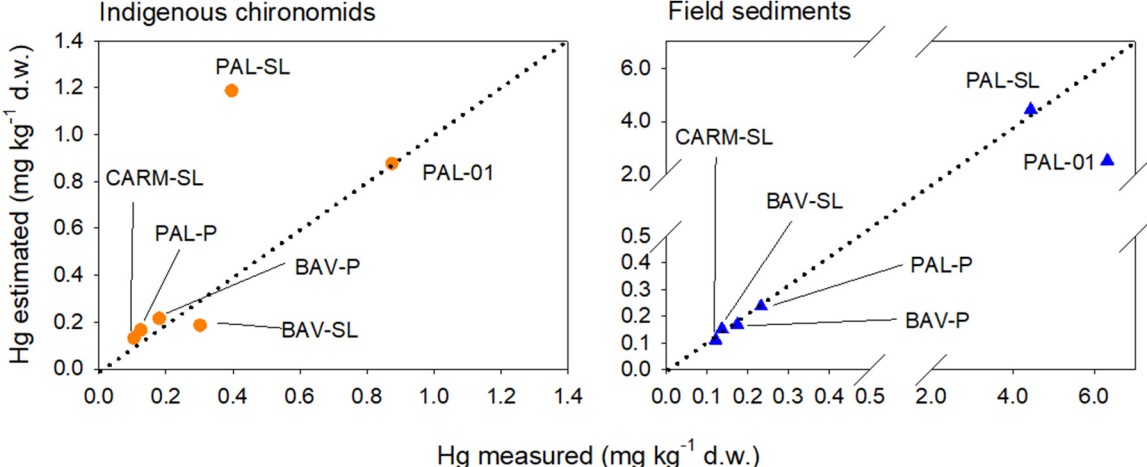

**Figure 3.** Scatterplot of measured vs. estimated Hg concentration values for 'Indigenous chironomids' and 'Field sediments'. Estimated values were obtained using regressions 1 and 2 (see text) for chironomids and sediments, respectively. The dashed lines indicate the ideal 1:1 correspondence.

With respect to these outliers, a limitation of the present study is that the variability of Hg accumulation in indigenous organisms was not determined because only one composite sample per site could be obtained to reach useful quantities of organisms to carry out the analyses despite the significant sampling effort.

Similar considerations apply to laboratory bioaccumulation tests with *C. riparius*, in which organisms from replicate exposures had to be pooled before analysis due to their limited mass. On the other hand, laboratory experiments with *L. variegatus* allowed for replicated analysis (Table S1) and showed a low inter-replicate variability (less than 30%) in most albeit not all cases. Assuming a 30% variability for replicate bioaccumulation measurements, the differences in Hg accumulation between indigenous chironomids and laboratory-exposed *C. riparius* for site PAL-SL (Figure 3) would not be solely controlled by variable Hg levels in different sediment aliquots. Sediment freezing to kill the indigenous fauna before laboratory tests may have modified Hg bioavailability before exposure of laboratory organisms and should be verified as a possible confounding factor. Furthermore, changes in the availability of sediment-bound elements between field and laboratory conditions have been documented [15] and may be valid in the case of mercury. These factors probably contributed also to the absence of correlation in Hg bioaccumulation by indigenous oligochaetes and *L. variegatus*. Calculation of BSAF takes into account changes in Hg sedimentary content and suggests that, except for site BAV-SL, the bioavailability of sediment-bound Hg to chironomids was higher in laboratory conditions than in the field, while the situation was more variable for oligochaetes (Table 2). In the case of accumulation by *L. variegatus*, some replicate measurements were highly variable (Table S1) and could affect BSAF estimation based on the average concentrations (Table 2).

The actual species composition of indigenous assemblages (Table S5) is another factor probably contributing to differences in Hg bioaccumulation between indigenous and laboratory organisms. In general, oligochaetes represented from 52% to 94% of the annual total absolute abundance at all the surveyed stations. Diptera chironomids were the second most abundant (1%–43% of the entire assemblage) and diverse taxon, followed by bivalves (1%–5%). The remaining taxa groups represented an insignificant part of the benthic community (0.2%–1.2%) and can be considered rare (Table S5). In detail, PAL-SL has the highest biodiversity in term of number of taxa found (35 taxa compared to

the other stations ranging between 14 and 29 taxa) and also the highest number of detritivorous taxa (18 taxa compared to other stations ranging between 7 and 16 taxa). However, BAV-SL showed the highest density for detritivorous chironomids (and the second highest for oligochaetes—Figure 4), while BAV-P had the highest density of detritivorous oligochaetes (Figure 4). These high densities suggest the presence of nutrient enrichment at the two stations.

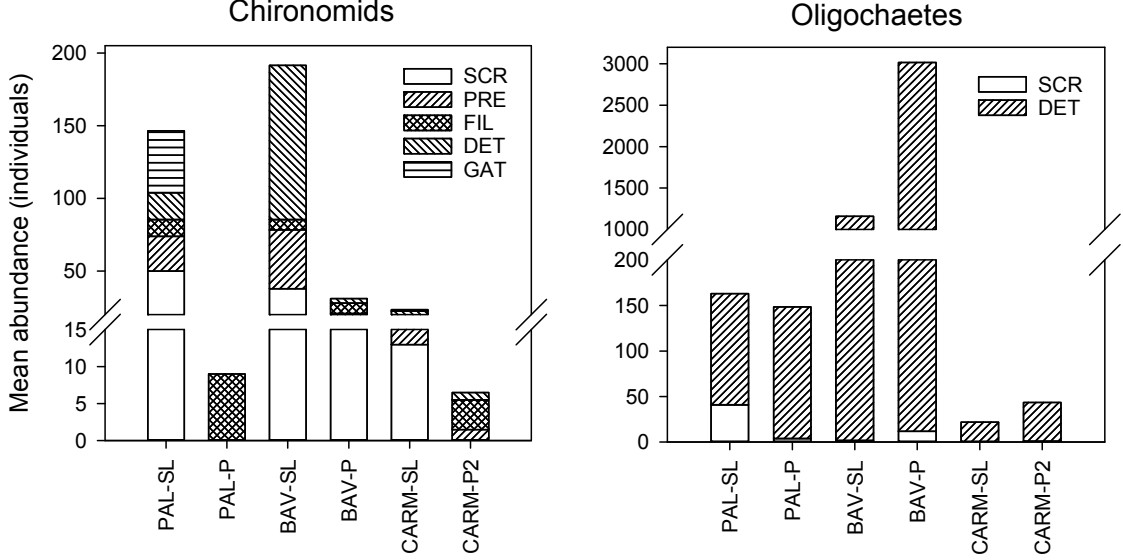

**Figure 4.** Annual mean densities of chironomids and oligochaetes according to the respective functional feeding groups as defined by USEPA (https://archive.epa.gov/water/archive/web/html/app_b-7.html). SCR: scraper; PRE: predator; FIL: filterer; DET: detritivores; GAT: gatherer.

The feeding habits of indigenous oligochaetes and chironomids as a whole (Figure 4) are not necessarily those of *C. riparius* and *L. variegatus*, which could result in different Hg accumulation patterns for indigenous and laboratory organisms. While indigenous detritivorous oligochaetes were always the predominant group at all the surveyed stations, detritivorous chironomids (specifically, *C. anthracinus* gr.) were the most abundant feeding group only at BAV-SL (Figure 4).

*Chironomus anthracinus* ingests particulate matter freshly deposited at the sediment surface [55]. Mercury bioaccumulation in this species likely reflects Hg content at the sediment–water interface with a degree of vertical selectivity that cannot be captured in laboratory tests with homogenised sediment aliquots. Conversely, the scraper *Prodiamesa olivacea* and the gatherer *Pagastiella orophila* were the most abundant species at PAL-SL (Table S5). The latter species has been shown to feed selectively on pennate diatoms [56], which could be an additional source of Hg not available during laboratory experiments. Furthermore, individuals of *C. riparius* analysed for Hg at the end of laboratory exposure were all in the fourth larval instar, while indigenous chironomids from the various taxa most likely were at different stage of development. The importance of feeding habits in interpreting differences in Hg bioaccumulation between indigenous oligochaetes and *L. variegatus* is less clear considering that detritivorous oligochaetes predominated at all sites (Figure 4 and Table S5).

## 4. Conclusions

Standardized laboratory tests with the model organism *C. riparius* suggested the possibility to predict Hg bioaccumulation in indigenous chironomids collected from a deep subalpine lake suffering from legacy Hg pollution. On the other hand, analogous experiments with *L. variegatus* as a proxy for indigenous oligochaetes were inconclusive. Factors causing differences between laboratory and indigenous organisms can include temperature differences between field versus laboratory settings, patchy sediment Hg contamination, variable degrees of Hg methylation and actual composition and developmental stages of indigenous chironomids and oligochaetes assemblages. Some factors (e.g.,

degree of Hg methylation) would be addressed easily from a technical point of view, while others would require in situ experiments that are, at best, logistically challenging and, for the moment, impracticable in deep lakes or in large, regular programmes.

Despite the mix of positive and negative results of the present study, laboratory bioaccumulation tests, therefore, retain their interest as tools in risk-assessment programs, although they should not be used alone for decision-making purposes. Indeed, sediment ecotoxicity tests are completely uninformative as to the potential for Hg remobilization from sediments, which is the first step to evaluating the risk for potential biomagnification in food webs. In this sense, the test with *C. riparius* is extremely versatile as it allows the assessment of ecotoxicity and bioaccumulation in both larval and adult (aerial) organisms using a single experiment. The tests with *C. riparius* and *L. variegatus* are also ecologically complementary thanks to the different feeding habits of the two model species. However, sediment sampling and handling for the preparation of laboratory tests can mix sediment layers with different Hg content. Collection of unperturbed sediments using coring devices instead of a grab sampler could limit the extent of such problems, improve the predictive ability of laboratory tests (including evaluation of different routes of uptake), and fully exploit the complementarity between different model organisms.

**Supplementary Materials:** The following are available online at http://www.mdpi.com/2076-3417/10/6/1970/s1: Table S1: Raw data for total mercury concentrations in *Lumbriculus variegatus*, Table S2: Total mercury concentrations in sediment cores LM16, LM17 and LM51, Table S3: Mercury content in surface and subsurface layers of sediments used for the exposure of *Lumbriculus variegatus*, Table S4: Total Hg concentrations in adult chironomids, Table S5: List of species of chironomids and oligochaetes, Figure S1: overview of laboratory bioaccumulation tests procedures, Supporting text: demonstration of negligible contribution of indigenous organisms to Hg sediment content.

**Author Contributions:** Conceptualization, D.A.L.V., R.B., A.B., S.V.; Validation, D.A.L.V.; Formal analysis, S.V.; Investigation, D.A.L.V., R.B., A.B., S.V.; Writing—original draft preparation, D.A.L.V.; Writing—review and editing, R.B., A.B., S.V.; Visualization, A.B.; Supervision, R.B. All authors have read and agreed to the published version of the manuscript.

**Funding:** This research was partially funded by ARPA Piemonte–Dipartimento del VCO di Omegna (VB, Italy).

**Acknowledgments:** The help of Giorgio Grilli in performing the laboratory tests with oligochaetes and of Annalisa Pola in Hg analyses is greatly acknowledged. We are indebted to the personnel of ARPA Piedmont, Omegna, Italy for their support in field activities.

**Conflicts of Interest:** The authors declare no conflict of interest.

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
