# Peer review of "Testing the Use of Standardized Laboratory Tests to Infer Hg Bioaccumulation in Indigenous Benthic Organisms of Lake Maggiore (NW Italy)"

_applsci, doi:10.3390/app10061970_

Round 1
Reviewer 1 Report
The authors focused their research on a very interesting topic, that is “establishes an environmental quality standard for Hg based on the total Hg concentration in prey fish to protect top predators”. I believe that work can be of great interest to researchers working in this field.
The work is well written and well organized; the adopted methods are adequately described, and the results are well presented.
I feel that this manuscript could be considered for publication after minor revisions.
Only one question: How the sediments samples have been stored before the treatment?
I suggest to add a section “4. Conclusions” and to improve the conclusions to underline the importance of the obtained results.
Author Response
Please see attachment: cover letter and responses to reviewer 1.

Reviewer 2 Report
The manuscript deals with the assess Hg bioaccumulation in indigenous and laboratory-reared chironomids and oligochaetes. Indigenous organisms were collected at different sites in a Lake Maggiore, whereas laboratory organisms were exposure to field-collected sediments. The Authors presented a large number of references to create the initial literature review and the results section. However, article has serious flaws: research is not design correctly, results are not interpreted appropriately and are not significant to predicting Hg bioaccumulation, article is not well written, it has no logical flow and applied methodology and obtained results are quite hard to understand. For these reasons, I cannot recommend this manuscript for publication.
The introduction sufficiently shows background and problem of Hg bioaccumulation is placed in a broad context. However, novelty of this work is not emphasized at all.
Lines 66-73: This fragment must be improved to in logical and evident way shows the originality of this manuscript and possibility of potential application.
Materials and methods: this part of manuscript is written in extremely incoherent way and do not allow another researcher to reproduce this experiment.
Lines 88-90: Authors wrote that samples were collected in 2010, but analysis of Hg for Lake Mergozzo was carried out in 2011. Table 2 contains the Hg content in Lake Mergozzo in 2010 (as I suppose). This is unclear…Analysis was replicated? Why?
Lines 128-142: This section must be reorganized: three main purposes (lines 128-131) should be moved to introduction, lines 131-132 and lines 136-142 should be moved to part 2.3, because it concerns collection of organisms. Method of sediment collection must be described in more details.
Lines 133-135: “The number of grab samples could vary across sites…” Is it about sediment and organisms samples? In line 153 Authors wrote that were thee replicates…
Lines 145-147: If this methodology provides to take samples in spring and autumn, why samples were collected only in spring? Results obtained for one month do not give fully information about Hg bioaccumulation. Sampling should be repeated to check the variability per year and in each sites.
Lines 149, 153 and 161: How many sampling sites and samples were? 18 samples were for three replicates, but later Authors wrote that were two additional locations… Why and how many samples were collected in two additional sites? It is not explained.
Lines 160-161: Were organisms collected from two additional sites only for Hg analysis? Was not taxonomic identification performed?
Table 1: Abbreviation OC (%) is not explained.
Lines 193-194: Why sediments were combed through 2 mm sieve? As Authors mentioned in lines 316-319 it is the most likely cause of significance differences in Hg contents between field sediments and sediments used for laboratory tests. For this reason, comparison indigenous organisms with organisms exposed for sediments sieved though 2 mm sieve is questionable.
Lines 194, 203, 223, 227: Why Lake Maggiore water and spring water was used for test with C. riparius but reconstituted and deionised water was used for test with L. variegatus? When Lake Maggiore water was sampled and how it was prepared for tests? It is not described, though is important to results interpretation. Hg content in Lake Maggiore water should be determined.
Lines 197, 239, 258, 266, 267: Why tests were not carried out in equivalent series? How many replicates were? It is unclear… In line 197: “five replicate beakers”, but in line 258 “three experimental replicates” and in line 266 “two exposure replicates”.
Lines 201, 227: C. riparius were fed but L. variegatus were not fed. It has a huge impact on obtained results, because added food contains Hg (as Authors mentioned in line 420, but they did not explain why they applied different feeding conditions).
Lines 227, 246: This information is repeated unnecessarily.
Lines 203, 204, 242-245: Why these parameters were measured daily for L. variegatus, but for C. riparius at the beginning and at the end of test only?
Lines 256, 257: How sediment samples were prepared for analyses? As Authors mentioned in line 191: “sediments samples were frozen at -20°C to eliminate indigenous macrofauna”. If sediments were analysed after freezing, obtained results Hg contents in samples sediment are burden with Hg from detritus.
Results (and Discussion): Interpretation of results must be carefully improved.
Table 2: Why results of Hg contents for ChInd, ChLab, Oliind were presented without standard deviations?
Lines 370, 399: For these wide ranges of results calculation of means is not appropriate. In these cases comparison of means has not statistical significance.
Lines 275, 407: This information is repeated unnecessarily.
Lines 371,372, 400, 401, 481, 482: Authors very inconsequently interpret Spearman correlation coefficient: “…weakly correlated…rho = 0.75…rho = 0.619…rho = 0.321…”, “…no correlation existed…rho = 0.68…”, “…but not for oligochaetes…rho = 0.43…”.
Research must be repeated, because predicting Hg bioaccumulation based on 18 samples (??? Amount of samples is unclearly described in manuscript…), which were collected at 8 sites (if I well understood) only in one month, is extremely doubtful. Additionally, laboratory tests and results interpretation were carried out in ways, which does not allow to draw clear conclusions.
Author Response
Please see attachment: cover letter and responses to reviewer 2.

Reviewer 3 Report
The manuscript "Predicting Hg bioaccumulation in indigenous organisms of Lake Maggiore (NW Italy) through standardized laboratory Tests" by Davide Vignati et al. aims to the present study evaluated Hg accumulation in assemblages of indigenous chironomids and oligochaetes could be predicted using standardized laboratory bioaccumulation tests with Chironomus riparius and Lumbriculus variegatus. The topic of the study fits the scope of the Journal of Applied Sciences and the article will be interesting for specialists in this research area as it provides new original data about field-collected sediments were analyzed for Hg content and laboratory bioaccumulation tests at Hg accumulation in chironomids and oligochaetes.
Results presented in manuscript are interesting, however, the manuscript needs to be revised and corrected before publication.
In Table 1 and Table 2, authors should reduce and rewrite the captioning, as they too long and very confusing.
Indicate what methods mercury was determined in sediments and in the tissues of organisms.
The goal set in the research must be approved in the Сonclusion. In this manuscript, the conclusion does not describe the results obtained by the authors. The authors should present their Conclusion in a separate section.
Discussion part is strongly needed. In general the authors should discuss the results and not only report it. Use more reference to the literature when discussing your research results.
Author Response
Please see attachment: cover letter and responses to reviewer 3.

Reviewer 4 Report
The paper „Predicting Hg bioaccumulation in indigenous organisms of Lake Maggiore (NW Italy) through standardized laboratory tests” fits the scope of the journal. Problem of environmental pollution by mercury is currently one of the major issues studied in numerous scientific centers due to the high Hg toxicity to living organisms. However, the description of the experimental part is chaotic, there are numerous contradictions and ambiguities in the text. I could not find the novelty of this paper. Therefore, I do not recommend publishing the work in its current form.
Below are examples justifying my decision:
Introduction – authors should briefly and clearly mention the main aim of the work, highlight the main conclusions and emphasize the novelty of their work (according to the instructions for authors). Chapter 2.1. lines 84-91 - the description of the sampling sites is not chronological. As far as I understand, in 2010 sediments were collected from Lake Maggiore (or in 2011 – Table S5?), and the following year - from Lake Mergozzo. Moreover, the mercury content results in Lake Mergozzo should be moved to Chapter 3. Chapter 2.1. – subsection title inadequate - no sampling description in this section. Chapters 2.2. and 2.3. - statements in lines 133-135 (“The number of grab…”) and 153 (“Each transect…”) are inconsistent. It should be specified the number of grab taken for different purposes. Chapter 2.4. lines 187, 194, 203, 213 and 227 – the information on tap water, lake, spring, reconstituted and deionised water parameters (as in lines 213, 214) are incomplete. Why such different types of water were used in both experiments? Lines 205, 206 – I understand that the purpose of purgation was to determine Hg bound in organisms, but it seems to me that if the purpose of the experiment was to assess the remobilization of Hg via benthic organisms, then the whole organism that top predators feed on should be analyzed (without purification). Chapters 2.4, 2.5 - different doses of food were used for each experiment, e.g. for the worms preparation: lines 217 - 100 mg of food twice a week; lines 230-231 control vessels - 150 mg 3 times a week. Does the diet not affect the condition of the test organisms? The same doubts apply to the way the overlying water is renewed in the vessels (e.g. lines 218-219 and 231-232). There is no information on separation from the sediment and analysis of native organisms. The work is not in accordance with the instructions for the authors (no "discussion" and "conclusions" chapters). Based on the data presented, the risk of mercury accumulation in the sediment for the food webs cannot be assessed. The authors do not consider the speciation forms of mercury in sediments. More research should be done at different times of the year.
Specific comments:
Table 1, 2, S4, Fig. 1, 2 – which code of sample Borromean Gulf is correct: PAL-01 or PL-01?
Table S5 – Pall_SL or Pal_SL? Pall_P or Pal_P? which code is correct?
Table 1 – what means superscript "1" (LM161, LM171 AND LM511) and what OC (%) means?
Literature is not in line with the journal's requirements
Author Response
Please see attachment: cover letter and responses to reviewer 4.

Round 2
Reviewer 2 Report
The authors have addressed each of comments. They reorganized the text, added important informations concern the methodology and explained doubtful parts of the research.The manuscript has been clearly improved.
Reviewer 4 Report
Dear Authors,
thank you for taking into account my comments. I think the new title fits the content much better. The corrections and additions added have made the description of the experience as well as the purpose of the work clearer. The authors have clarified a number of my doubts that have arisen in the previous version of the article. I also appreciate that you were critical of some of your results and the methodology used, and that you listed some of the weaknesses in the method. This justifies the need to continue research. Let me suggest that in the future a uniform research methodology should be agreed with cooperating institutes.
I can recommend publishing the work in its current form.
Specific comments:
Lines 227, 370 – please separate values and units
Lines 224, 230, 253, 293 – please remove spaces